# The Efficacy of Early Osteopathic Therapy in Restoring Proper Sucking in Breastfed Infants: Preliminary Findings from a Pilot Study

**DOI:** 10.3390/healthcare12100961

**Published:** 2024-05-08

**Authors:** Arianna Parodi, Rosalba Ruffa, Viola De Felice, Marina Sartini, Maria Luisa Cristina, Beatrice Martino, Francesca Bianco, Roberta Di Stefano, Massimo Mazzella

**Affiliations:** 1Neonatology Department, Ente Ospedaliero Ospedali Galliera, 16128 Genoa, Italy; rosarosanuova@gmail.com (R.R.); defelice.viola@gmail.com (V.D.F.); beatrice.martino@galliera.it (B.M.); bianco.francy@yahoo.it (F.B.); roberta.di.stefano@galliera.it (R.D.S.); massimo.mazzella@galliera.it (M.M.); 2Operating Unit Hospital Hygiene, Galliera Hospital, 16128 Genoa, Italy; marina.sartini@galliera.it (M.S.); maria.luisa.cristina@galliera.it (M.L.C.); 3Department of Health Sciences, University of Genoa, 16132 Genoa, Italy

**Keywords:** breastfeeding, newborns, neonatal, International-Board-Certified Lactation Consultant, Baby-Friendly Hospital Initiative, osteopathy, sucking reflex

## Abstract

Despite the care provided, some newborns, who are perfectly healthy, show functional alterations that impair a good breast attack in the first few days. This situation often leads to the early failure of lactation. We conducted a randomized single-blind controlled trial to evaluate whether four weeks of osteopathic treatment can normalize the sucking score in a group of neonates with impaired lactation ability. Forty-two healthy full-term neonates were enrolled in this study. On the basis of the sucking score and the assessment of the infant’s breastfeeding skills, infants who had intrinsic breastfeeding difficulties were selected. The inclusion criteria were healthy infants born > 37 weeks, a suction score ≤ 6, and any breast milk at enrolment. At the end of the study, the sucking score was significantly greater in the osteopathic group than in the untreated group; the median sucking score in the treated group was in the normal range, while it remained lower in the untreated group. At the end of the follow-up, the percentage of exclusively breastfeeding infants in the treatment group increased by +25%. This pilot study demonstrates the efficacy of early osteopathic intervention to relieve breastfeeding difficulties in newborns in the first few weeks of life.

## 1. Introduction

Breastfeeding is a globally recognized means for improving the health of babies and women. Around the world, several initiatives aim to support and sustain breastfeeding. Among organizations at the hospital level, Baby-Friendly Initiatives (BFIs) have the strongest scientific basis, demonstrating its effectiveness [1,2,3,4]. Baby-Friendly Hospitals (BFHs) provide a three-step validation procedure to demonstrate commitment to competent breastfeeding support. At the level of healthcare professionals, the International-Board-Certified Lactation Consultants (IBCLC) certification represents the highest level of expertise in breastfeeding assessment and in the implementation of strategies to support difficult breastfeeding, and it represents the gold standard for improving breastfeeding skills [5,6]. The setting of this study is a Baby-Friendly Hospital (BFH) where several IBCLCs work.

Despite the care provided, some newborns, who are otherwise perfectly healthy, show very early functional alterations that impair a good breast attack. This situation often leads to the early failure of lactation, mainly related to mother’s pain during latching but partly also due to problems related to a suboptimal milk transfer [7].

Breastfeeding pain is a major reason for the early cessation of breastfeeding for approximately 35% of women [8,9]. About 90% of women report acute breast and nipple pain during the first week of breastfeeding initiation, taking charge of this pain can help these women continue breastfeeding and prevent early cessation.

If there are difficulties in initiating valid suction patterns, it is necessary to understand how this mechanism occurs and what structures it uses. The breast suction reflex is much more complex and completely different from that of a bottle [10], and it is determined by several factors that begin to develop in utero and complete its function at the time of birth through lactation.

It is evident that the fetus, during intrauterine life, as well as in the progression and expulsion phase during childbirth, undergoes compression and stretching, particularly at the expense of cranial structures which are capable of altering harmony and balance among tissues and among the different bones, cartilages, and membranous components that build the neonatal skull and body [11]. Such situations can significantly impact the complex abilities of proper breast attack and suction [12]. An observational evaluation carried out at this facility showed that, at present, infants who were otherwise healthy but had an altered breast sucking pattern had lower exclusive breastfeeding rates at discharge than the general neonatal population did (45% versus 87.5%).

Osteopathic vision could provide a functional interpretation of the limiting factors of breastfeeding. In the neonatal period, the possibility of remodeling coating tissues through these functional osteopathic techniques is particularly favorable, and this technique is likely more effective the earlier it is taken care of [12]. Some studies and case reports have started shedding positive light on the effect of osteopathy on neonatal care [13]. Lund and Wescott [14,15] showed the possible effect of functional osteopathic procedures in the management of sucking dysfunctions. Herzhaft-Le Roy J et al. [16] reported a statistically significant improvement in latch scores in treated newborns.

Other studies, such as that by Marie Danielo Jouhier [17], have shown discordant results.

To explore this topic, which is much debated among those involved in breastfeeding, we decided to start a single-blind randomized study with the primary objective of evaluating the impact of early osteopathic treatment on the resumption of valid breast sucking in a population of infants with incorrect sucking patterns.

## 2. Methods

### 2.1. Research Design

This was a randomized single-blind controlled trial with a 1:1 allocation ratio on two treatment arms (standard vs. standard + osteopathic treatment). The primary aim was to evaluate whether four weeks of osteopathic treatment can normalize the sucking score in a group of neonates with impaired lactation ability.

The secondary aim was to evaluate breastfeeding rates after treatment in the two groups. and to understand if improving the sucking score also improves the prevalence of breastfeeding.

### 2.2. Participants

In this observational study, 42 healthy full-term newborns were enrolled.

The inclusion criteria for healthy infants were an age > 37 weeks and early impaired latching in the first 48–60 h of life (a suction score ≤ 6).

The exclusion criteria were infants born before 37 weeks of gestation, infants who recovered from neonatal intensive care units (NICUs), infants with facial malformations, maternal breasts/nipples that make latching difficult, and formula feeding for maternal choice. All infants who met the inclusion criteria (born between January 2021 and January 2023) were enrolled, with interruptions related to the progress of the COVID-19 pandemic in Italy.

### 2.3. Setting

This study was conducted at a Baby-Friendly Hospital in northwestern Italy. In our facility, approximately 900 neonates are born per year, and approximately 85% of the newborns are fully breastfed at discharge.

### 2.4. Method

A team of lactation consultants (IBCLCs) evaluated infants with breastfeeding difficulties in the first two days of life. Suckling in newborns has been evaluated through the use of a sucking score, which allows us to determine which areas are altered in an effective and repeatable way [18]. Scores ranging from 0 to 2 were assigned for cupping, the position of the tongue, the movement of the tongue, and the movement of the jaw.

To find the sucking score, the observer assessed if the tongue posture (observed through lower lip lowering and, if necessary, jaw lowering) was flat, elevated, retracted, or protruded; if non-nutritive sucking tongue movement was adequate, altered, or absent; if tongue cupping was present or absent; and, finally, if jaw movement was adequate, altered, or absent. Normal sucking was defined as a score above 6. A total score ≤ 6 indicates an altered sucking pattern [18].

On the basis of the score and the assessment of the infant’s breastfeeding skills, infants who had intrinsic breastfeeding difficulties were selected and subsequently enrolled in the study and randomly allocated two different arms of treatment.

The two study arms are as follows:-Control group: Infants who received “standard care”, performed by an IBCLC team that deals with difficult breastfeeding, in a setting that routinely implements the ten steps of the Baby-Friendly Initiative;-Treatment group: In addition to the standard treatment provided to the control group, infants will undergo weekly treatment for the first 4 weeks of life (Tr1–Tr4) via a dedicated osteopath.

At the time of randomization (T1), infants in both groups were evaluated by an osteopath (other than the one performing the treatment who acted in a blinded manner and exclusively assigned the osteopathic assessment score). In the middle of the intervention time (T2) and at the end of the treatment (T3), all infants were evaluated to define the assignment of lactation classes according to the WHO/“Baby-Friendly” criteria (exclusive, predominant, mixed feeding, and artificial) and osteopathic and sucking scores (see Figure 1: Experimental flow chart (Tr = osteopathic treatment)).

All the infants’ families provided signed informed consent for these data to be used for research purposes. The Provincial Ethics Committee of Liguria approved the study (N. Registro CER Liguria: 292/2020-DB id 10288), both in terms of the type of intervention and in terms of the privacy and storage of the sensitive data of our patients. The study was entered into EudraCT 2019-001007-20 “OsteoNeo”.

For every recruited patient, maternal, perinatal, and breastfeeding data were collected. The data were collected anonymously on Excel databases. The quality of the data was assessed by two researchers.

### 2.5. Osteopathic Intervention (Evaluation and Treatment)

The osteopathic evaluation aimed to assess the presence of osteopathic dysfunctions in structures that may have an impact on sucking. At the end of the evaluation, all the neonates received an “osteopathic score”. The osteopathic evaluation score, in the absence of validated scores in the literature, was created ad hoc by Dr. Ruffa R. before participant enrollment, and a specific osteopathic evaluation was designed and described as follows.

The osteopathic evaluation of a neonate, in addition to the anamnesis, provides the information necessary to formulate the diagnosis of dysfunction. Somatic dysfunction is defined as a “restriction of tissue mobility” and encompasses hemodynamic, neurovegetative, hormonal, and humoral alterations that affect the whole organism. The severity of an injury is defined by the slowness, absence, or restriction of the movement of the tissues, considering all the examined parts, even those that are remotely involved. The palpatory assessment allows the osteopath to identify areas of greater density or tissue impairment with contact as light as possible to thoroughly unfold the anatomic shape of the part under examination without affecting it. After the evaluation, functional osteopathic treatment relieves the tensions encountered. Functional osteopathy is a holistic approach to healthcare that focuses on the interrelationship between the structure and function of the body. When applied to newborns, functional osteopathy aims to address any potential imbalances or dysfunctions that may arise during birth or in the early stages of life. Functional osteopathy techniques for newborns are gentle and non-invasive, typically involving subtle manipulations of the baby’s body. These techniques may include gentle stretches, massages, and positional release techniques aimed at releasing tension in the muscles, joints, bones, and connective tissues. Practitioners may also work to optimize the baby’s cranial and spinal alignment to support optimal nervous system function.

### 2.6. Statistical Analysis

All patient characteristics are presented as the means, standard deviations, medians, and ranges for continuous variables and as absolute values with percentages for categorical variables. Due to the absence of a normal distribution in the data, various numerical transformations were considered. However, none of these transformations effectively mitigated the skewness observed. Consequently, nonparametric tests were employed for data analysis. The Kruskal–Wallis test was utilized for continuous variables and the chi-squared test was utilized for categorical variables to assess the independence between the variables. To compare sucking scores before and after the intervention in each of the two distinct groups, the Wilcoxon matched-pairs signed-rank test was employed. All tests were two-sided, and a *p* value less than 0.05 was considered to indicate statistical significance. The statistical analyses were conducted using Stata/SE 18.0 software (StataCorp LP, College Station, TX, USA).

Data management: Data were recorded in an electronic database that guaranteed the blinding of the outcome assessor.

## 3. Results

In total, 48 mother–baby dyads were enrolled, and 6 dyads were lost to follow-up due to the intercurrent illness of the newborn or mother. Eventually, each arm was composed of 21 pairs.

The characteristics of enrolled mothers are reported in Table 1; no statistically significant differences were observed for the variables considered between the standard group and the treated group.

Both groups showed similar features at T1 in terms of gestational age, BAS score, type of delivery, parity, and sucking score. At T1, the intervention group had more complementary breastfed babies than the standard group did (see Table 1). At the end of the study, the group of treated infants showed a significantly greater normalization of the sucking score than the group of untreated infants did. The median score in the treated babies ranged from 3 (3.67 ± 1.06) at T1 to 6.5 (6.35 ± 1.27) at T3, while in the control group, it ranged from 4 at T1 (3.86 ± 0.91) to 5 (5.45 ± 1.05) at T3, with *p* < 0.001 (see Figure 2: Sucking scores in the two groups at T1 and T3). Change was significant for both groups, but the increase was greater in the treated group. The final score in treated patients was significantly higher than in the untreated group (*p* < 0.001). In other words, the sucking score in treated babies veered towards normality compared to untreated babies, in whom the lack of suction remained and mostly had pathologic values (see Figure 2).

By time T3, the percentage of infants who exclusively breastfed in the treatment group increased by +25%, while that in the standard group decreased by −10% (see Figure 3), even if the difference did not reach statistical significance.

The osteopathic score decreased more significantly in the treated group than in the standard group. By T1, the median score was similar in both groups (standard 77.0 vs. treated 80); at T3, the median osteopathic score was 47 in the standard group and 15 in the treatment group (*p* < 0.001). At the halfway point of the study, after 15 days, the difference between the two groups was statistically significant (a median standard score of 62 vs. a median treatment score of 46, with *p* < 0.001) (Table 2).

## 4. Discussion

Breastfeeding is a natural and invaluable gift that not only nourishes the body but also nurtures the bond between mother and child while providing a solid foundation for lifelong health and well-being. It is more than just a feeding method; it is a cornerstone of early childhood development with a myriad of benefits. Improving early breastfeeding difficulties quickly and decisively ensures the possibility for the mother and child dyad to be able to continue breastfeeding. Numerous studies have shown that difficulties encountered in the first week led to the early termination of breastfeeding [7,8,19].

The skills of health workers, even if they are based on the utmost professionalism, as in the case of IBCLCs, do not always guarantee this goal.

On the basis of previous studies, functional osteopathy is a particularly favorable technique used for solving various problems that may arise in neonates [11,13,20,21].

Functional osteopathy in newborns focuses on addressing any strains, tensions, or restrictions that may have occurred during the birthing process. These strains or tensions can manifest in various ways, such as breastfeeding difficulties, colic, reflux, or sleeping issues. Practitioners of functional osteopathy utilize gentle hands-on techniques to assess and manipulate the musculoskeletal system, as well as other bodily systems, such as the nervous system, the lymphatic system, and the circulatory system.

Position during intrauterine life, as well as the progression and expulsion phase during childbirth, particularly at the expense of the cranial cavity, seems to be capable of altering harmony and balance among tissues, bones, cartilage, and membranous components that build the neonatal skull. In some circumstances, both normal and abnormal labor mechanics are considered potential traumatic factors for the craniosacral mechanism [19]. The compressive forces that the newborn undergoes during childbirth remain enclosed in the tissues and restrict the intrinsic movement of the tissues themselves. Sometimes, the mechanism of childbirth is already affected by the intrauterine position of the fetus. This aspect is fundamental for understanding that somatic dysfunction may have an embryonic or postnatal origin. Such dysfunctions can significantly impact the complex abilities of proper breast attack and suction in otherwise healthy newborns. In this study, randomization was effective, and no difference in the type of delivery was observed between the two groups.

A randomized controlled trial by Martelli [22] investigated osteopathic sham therapy and showed that no placebo effect occurs in newborns. Previous work [11,21] has shown that dysfunctional patterns in the skull have considerable importance in the development of symptoms such as vomiting, peristaltic hyperactivity, tremor, hypertonia, and irritability. The osteopathic scenario, however, lacks standardized guidelines for the evaluation and treatment of newborns.

We therefore looked for a more holistic approach that would allow us to help complex cases that do not benefit from a standard approach.

Our work differs in the precocity of the intervention and in the setting within a BFH hospital where the gold standard of care was already ensured for our patients, as all latch difficulties were addressed by an experienced IBCLC team and all ten steps were implemented.

At the end of this pilot study, the treated group showed an improvement in the sucking score and a greater reduction in the osteopathic score than the untreated group (*p* < 0.001).

It should be noted, however, that the infants in the control group also showed an improvement in performance. We hypothesized that “training” related to mouth movements during breastfeeding is capable of bringing small improvements in itself. This once again underlines the fundamental importance that breastfeeding has on all-round health, including on the function/structure of the mouth.

Differences in terms of breastfeeding rates were not significant, but the trend was positive. We can assume that the lack of significance could be related to the relatively small number of patients and the fact that variables at play speaking during breastfeeding are not always easy to discriminate. For example, mothers’ motivation to breastfeed despite difficulties has not been fully investigated. Moreover, in our setting, breastfeeding support, even in the standard group, was already high. One would imagine that in a different setting, i.e., in a non-BFH setting, the breastfeeding results could have been more remarkable.

In this pilot paper, we showed that healthy newborns who are difficult to breastfeed can have varying degrees of osteopathic dysfunction. In conclusion, early functional osteopathy assessment techniques may be helpful in detecting such dysfunctions, and these preliminary data confirm that early osteopathic treatment may be effective for improving the resilience of sucking modalities in otherwise healthy newborns.

### Limitations

Our study has its limitations. Only demographic data and perinatal information were included, and the factors influencing individuals’ attitudes and behaviors were not included. Thus, we suggest that future studies include more possible influencing factors, such as social support, postpartum mood, self-efficacy, and motivation for breastfeeding. We achieved a statistically significant difference in the number of patients who sucked, which was lower than expected; it is likely that, with a greater number of patients, we can also observe a significant difference in terms of breastfeeding.

Finally, this was a preliminary pilot study in which an osteopathic score that was created ad hoc and not validated was used, as validating this score is not the purpose of this pilot study. A subsequent investigation on a larger sample of patients could allow to validate the score. More patients are also needed to clarify the potential benefit of functional osteopathy in newborns with breastfeeding issues. Eventually, a larger number of patients is also needed to better analyze the role of sucking score in breastfeeding performance.

## 5. Conclusions

This study is different from previous ones, especially due to the precocity of the intervention and the setting within a BFH hospital sucking score, as a marker of sucking impairment is also a novelty. In conclusion, this pilot study explores the impact of an early functional osteopathic therapy, started in the first week of life, on sucking problems related to breastfeeding in healthy infants.

## Figures and Tables

**Figure 1 healthcare-12-00961-f001:**
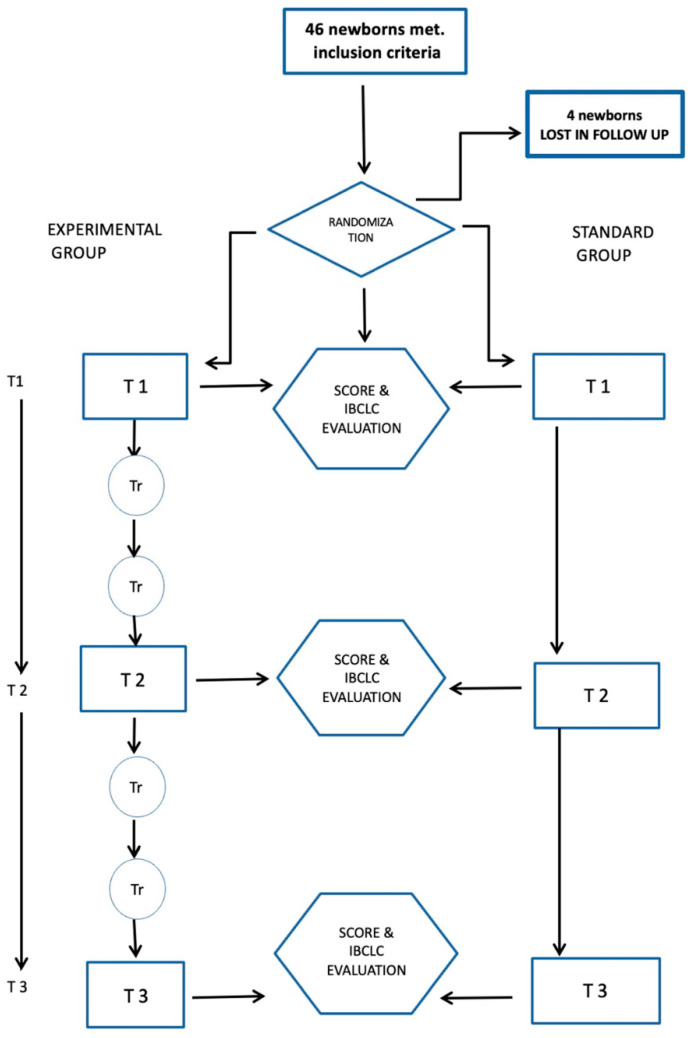
Experimental flow chart (Tr = osteopathic treatment).

**Figure 2 healthcare-12-00961-f002:**
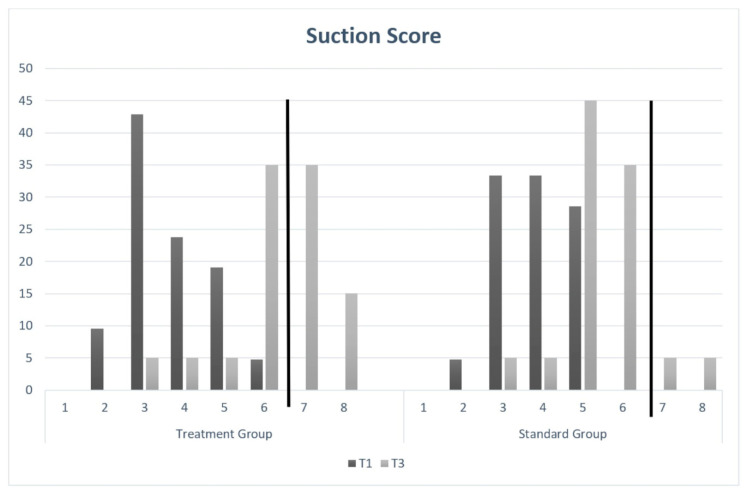
Sucking scores in the two groups at T1 and T3.

**Figure 3 healthcare-12-00961-f003:**
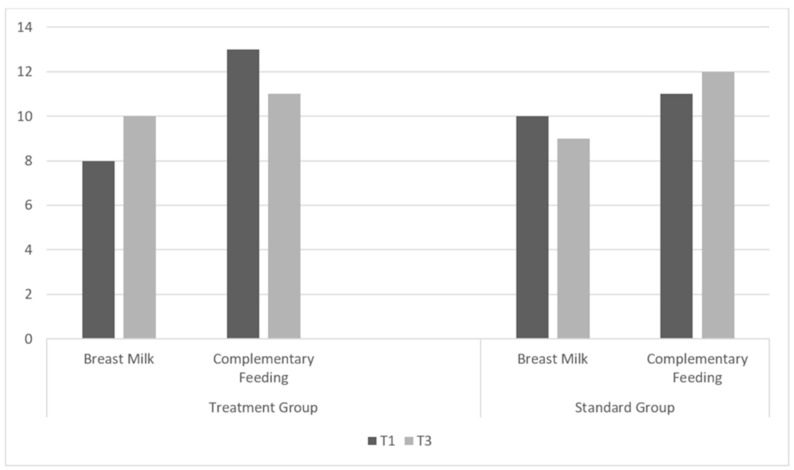
Number of children from any arm switched from mixed feeding (complementary feeding) to EBF.

**Table 1 healthcare-12-00961-t001:** Maternal, perinatal, and breastfeeding features.

		All	Standard Group	Treatment Group
Gestational age (weeks)	Mean ± SD Median Min–max	39 ± 2 40 30–42	39 ± 3 40 30–42	39 ± 1 40 39–40
BAS	Mean ± SD Median Min–max	4.5 ± 1.27 4 1–7	4.7 ± 1.52 5 1–7	4.28 ± 0.96 4 2–6
Suction score T1	Mean ± SD Median Min–max	3.76 ± 0.98 4 2–6	3.86 ± 0.91 4 2–5	3.67 ± 1.06 3 2–6
Osteopathic score T1	Mean ± SD Median Min–max	76.88 ± 13.51 78 52–102	77.00 ± 12.60 77 54–102	76.76 ± 14.69 80 52–102
Type of delivery	ND	27	12 (57.1%)	15 (71.4%)
	DD	5	3 (14.3%)	2 (9.5%)
	ECS	2	1 (4.7%)	1 (4.7%)
	UCS	8	5 (23.8%)	3 (4.7%)
Parity	Primipara	28	15 (71.4%)	13 (61.9%)
	Multipara	14	6 (28.5%)	8 (38.1%)
Type of feeding	EXCLUSIVE BM	18	10 (55.5%)	8 (44.4%)
	MIXED FEEDING	24	11 (45.8%)	13 (54.1%)

ND: normal delivery, DD: dystocic delivery, ECS: elective cesarean section, UCS: urgent cesarean section; BAS: breastfeeding assessment score, BM: breast milk.

**Table 2 healthcare-12-00961-t002:** Osteopathic score at T1, as well as in the middle (T2) and at the end (T3) of the study.

		All	Standard Group	Treatment Group	*p*-Value
Score osteo T1	Median ± SD Median min–max	76.88 ± 13.51 78 52–102	77.00 ± 12.60 77 54–102	76.76 ± 14.69 80 52–102	NS
Score osteo T2	Median ± SD Median Min–max	55.07 ± 16.99 54 24–102	63.95 ± 16.08 62 34–102	46.19 ± 12.94 46 24–68	<0.001
Score osteo T3	Media ± SD Median Min–max	33.6 ± 21.82 32 6–102	47.95 ± 21.22 47 10–102	19.3 ± 9.74 15 6–36	<0.001

NS not significant.

## Data Availability

The datasets in the current study are available from the corresponding author upon reasonable request.

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
