# Peer review of "The Efficacy of Early Osteopathic Therapy in Restoring Proper Sucking in Breastfed Infants: Preliminary Findings from a Pilot Study"

_healthcare, 2024, doi:10.3390/healthcare12100961_

Round 1
Reviewer 1 Report
Comments and Suggestions for Authors
Characterize what functional osteopathy is or what functional osteopathic procedures are used in the treatment of sucking disorders.
What were the focus of the weekly treatment interventions - complete the methodology
Further characterise the evaluation of the osteopathic intervention
Did the mothers of newborns enrolled in the study receive prenatal preparation?
Their skills related to breastfeeding positioning, breast holding, newborn handling were assessed.
What was the ratio of primipara to multipara?
Author Response
To the Editor and the Reviewers of Heathcare
About the paper “Efficacy of early osteopathic therapy in restoring proper sucking in breastfed infants” by Arianna Parodi, Rosalba Ruffa, Viola De Felice, Marina Sartini,Maria Luisa Cristina, Beatrice Martino, Francesca Bianco, Roberta Di Stefano, Massimo Mazzella
Arianna Parodi, as corresponding author, on behalf of all the authors, has made the requested changes so detailed
Characterise what functional osteopathy is or what functional osteopathic procedures are used in the treatment of sucking disorders.
I have added a paragraph to better characterise how functional osteopathy can be helpful in newborns
What were the focus of the weekly treatment interventions - complete the methodology
Aim of the osteopathic treatment is to improve the osteopathic dysfunctions underlying the newborn's sucking problems
Further characterise the evaluation of the osteopathic intervention
I have added a paragraph to better characterise how functional osteopathy can be helpful in newborns
Did the mothers of newborns enrolled in the study receive prenatal preparation?
All women that deliver in Galliera Hospital receive information about breastfeeding during pregnancy as the third step of Baby Friendly Initiative (BFI) recommend
Their skills related to breastfeeding positioning, breast holding, newborn handling were assessed.
All women that deliver in Galliera Hospital receive information about breastfeeding positioning, breast holding and newborn handling as recommended by the 5th step of BFI
What was the ratio of primipara to multipara?
I have added the ratio of primipara to multipara in table 1, no statistical significance was found in the two groups about parity
Reviewer 2 Report
Comments and Suggestions for Authors
Thank you for the opportunity to review this interesting intervention pilot study. The authors address an important topic: The efficacy of early osteopathic therapy in restoring proper sucking in breastfed infants. My feedback is as follows:
Title: I suggest adding the research design in the title to become “Efficacy of early osteopathic therapy in restoring proper sucking in breastfed infants: A preliminary findings from a pilot study.
Abstract:
All abbreviations should be defined the first time they appear in three sections: the abstract, the main text, and the first figure or table).
No Keywords are present in the document (Three to ten pertinent keywords are needed)
1. Introduction:
- According to the journal style: the first section is the introduction. Therefore, please add 1. Introduction instead of 1. Early osteopathic evaluation in healthy neonates with impaired 22 breastfeeding competence
- More clarification is needed to address the osteopathic therapy in restoring proper sucking in breastfed infants
- The authors must address the gap between the current study and the previous studies and highlight the topic's novelty.
2. Methods
- There are redundancies in the inclusion and exclusion criteria, such as” the exclusion criteria for infants were infants born before 37 weeks of 85 gestation”.
- More details are needed to describe the Assessment scale of newborn sucking for breastfeeding and other data collection instruments such as demographic data, maternal, perinatal, and breastfeeding features.
- How did the authors recruit their subjects (Data collection procedures and quality control)? What are the ethics considered when conducting this study? I can't find the answers to these questions mentioned.
- More details are needed to clarify functional osteopathic intervention that relieves the tensions after newborn evaluation.
3. Results
- Table 1. Data about maternal, perinatal, and breastfeeding features. It should be transferred to the results section.
- Figure 3. Without title
- Table 2: the head of the table was not found and I can’t able to confirm the difference between the standard group and the treatment group.
4. Discussion
The discussion required more depth in the interpretation of the findings.
5. Conclusion: where????
Best Regards
Comments on the Quality of English Language
Moderate editing of English language required
Author Response
To the Editor and the Reviewers of Heathcare
About the paper “Efficacy of early osteopathic therapy in restoring proper sucking in breastfed infants” by Arianna Parodi, Rosalba Ruffa, Viola De Felice, Marina Sartini,Maria Luisa Cristina, Beatrice Martino, Francesca Bianco, Roberta Di Stefano, Massimo Mazzella
Arianna Parodi, as corresponding author, on behalf of all the authors, has made the requested changes so detailed
Thank you for your suggestion!
Title: I suggest adding the research design in the title to become “Efficacy of early osteopathic therapy in restoring proper sucking in breastfed infants: A preliminary findings from a pilot study.
- As suggested the title has been changed to “Efficacy of early osteopathic therapy in restoring proper sucking in breastfed infants: A preliminary findings from a pilot study”
Abstract:
All abbreviations should be defined the first time they appear in three sections: the abstract, the main text, and the first figure or table).
No Keywords are present in the document (Three to ten pertinent keywords are needed)
- the abstract has been corrected by explaining the abbreviations
- Keywords have been added, they were put in a separated file named “title page”
Introduction:
- According to the journal style: the first section is the introduction. Therefore, please add Introduction instead of 1. Early osteopathic evaluation in healthy neonates with impaired 22 breastfeeding competence
- More clarification is needed to address the osteopathic therapy in restoring proper sucking in breastfed infants
- The authors must address the gap between the current study and the previous studies and highlight the topic's novelty.
- First section of the Introduction has been corrected
- As suggested also by other reviewers a paragraph better explaining functional osteopathy has been added
- We have tried to better underline how the present study is different from the previous ones, especially thank to the precocity of taking charge of sucking difficulties. Sucking score as a marker of sucking impairment is also a novelty
Methods:
- There are redundancies in the inclusion and exclusion criteria, such as” the exclusion criteria for infants were infants born before 37 weeks of 85 gestation”.
- More details are needed to describe the Assessment scale of newborn sucking for breastfeeding and other data collection instruments such as demographic data, maternal, perinatal, and breastfeeding features.
- How did the authors recruit their subjects (Data collection procedures and quality control)? What are the ethics considered when conducting this study? I can't find the answers to these questions mentioned.
- More details are needed to clarify functional osteopathic intervention that relieves the tensions after newborn evaluation.
- Exclusion criteria have been corrected
- A paragraph to better explain how sucking score is performed has been added in Method section
- Ligurian Ethic Committee has approved the present study, as stated at the end of method section
Results
- Table 1. Data about maternal, perinatal, and breastfeeding features. It should be transferred to the results section.
- Figure 3. Without title
- Table 2: the head of the table was not found and I can’t able to confirm the difference between the standard group and the treatment group.
- Data about maternal, perinatal and breastfeeding features has been moved in the Result section
- Figure 3 title and Table 2 head have been modified as requested
Discussion
The discussion required more depth in the interpretation of the findings.
- Conclusion: where????
- Conclusion paragraph and a better characterization of the findings have been added
Reviewer 3 Report
Comments and Suggestions for Authors
Abstract: 1) Remove the subheadings, 2) Did you use any statistical test for assesing the difference between control and test group? Write the name.
What is the full form of IBLCL.
The diagram is very light. I cannot see it properly. Please make a clearly visible diagram.
I do not believe for the introduction of complementary feeding in children aged below 2 months. It could be predominant feeding or supplemental feeding. Check this.
Remove p-value column.
Write full form. It is extremely difficult to read the abbreviation of each and every category which you mentioned from Table-1.
Also mentioned percentages of different delivery methods and of feeding practices.
Can you please explain figure 3? This seems very confusing. You need to explain at T-1 how many children were on EBF from each arm, and how many were on CF on each arm. Then you need to show that the change with percentage. Moreover, you cannot say % of EBF increased or decreased at time T3. Rather you can say that the X% of children from arm X were switched to from CF to EBF. This sentence needs to be clear.
Discussion is not very coherent and concise. It need more citation. The external validity of study need to be stronger.
You need to add an additional heading of conclusion after limitations.
Author Response
To the Editor and the Reviewers of Heathcare
About the paper “Efficacy of early osteopathic therapy in restoring proper sucking in breastfed infants” by Arianna Parodi, Rosalba Ruffa, Viola De Felice, Marina Sartini,Maria Luisa Cristina, Beatrice Martino, Francesca Bianco, Roberta Di Stefano, Massimo Mazzella
Arianna Parodi, as corresponding author, on behalf of all the authors, has made the requested changes so detailed
Thank you for your suggestion!
Abstract: 1) Remove the subheadings, 2) Did you use any statistical test for assessing the difference between control and test group? Write the name.
- Abstract has been modified as requested, statistical test have been fully described in the method section
What is the full form of IBLCL.
- IBCLC has been modified in International Board Certified Lactation Consultant
The diagram is very light. I cannot see it properly. Please make a clearly visible diagram.
- Diagram of Figure 1 and Table 1 has been modified as requested
I do not believe for the introduction of complementary feeding in children aged below 2 months. It could be predominant feeding or supplemental feeding. Check this.
- We had used complementary feeding as WHO define any other nutrition besides human milk. As it may be confounding, complementary feeding has been changed in mixed feeding as suggested
Remove p-value column. Write full form. It is extremely difficult to read the abbreviation of each and every category which you mentioned from Table-1.
ok
Also mentioned percentages of different delivery methods and of feeding practices.
- Percentage of different delivery methods and of feeding practices have been added
Can you please explain figure 3? This seems very confusing. You need to explain at T-1 how many children were on EBF from each arm, and how many were on CF on each arm. Then you need to show that the change with percentage. Moreover, you cannot say % of EBF increased or decreased at time T3. Rather you can say that the X% of children from arm X were switched to from CF to EBF. This sentence needs to be clear.
- Explanation of results shown in Figure 3 has been bettered and changed as suggested
Discussion is not very coherent and concise. It need more citation. The external validity of study need to be stronger.
- Thank you for your suggestion, we think that citations in the Discussion section are coherent with our purpose, but we are very grateful if you can suggest others.
- External validity has not been taken into account because this is a small pilot study that assumes a subsequent investigation on a larger sample of patients
You need to add an additional heading of conclusion after limitations.
- Conclusion paragraph and a better characterization of the findings have been added
Reviewer 4 Report
Comments and Suggestions for Authors
Thank you for the opportunity to review the manuscript related to evaluate whether four weeks of osteopathic treatment can normalize the sucking score in a group of neonates with impaired lactation ability in Italy.
A few questions / comments and suggestions:
In introduction section lacks citations for some statements, such as in line 39-44, the claims that intrauterine position can alter neonatal skull bones and impact breastfeeding. Citations are needed to support these statements, relevant to the study is not clear.
In method section, in line 71 to 168, it states that an osteopathic score was created for evaluations, but details on how this score was developed, the components assessed, and validation are not provided. More information is needed on this part, relevant to the study is not clear.
In line 90 to 151, no exact duration and data for data collection, relevant to the study is not clear.
In result section, in line 169 to 197, statistical tests used to compare groups are not reported, so it is unclear if the reported differences are statistically significant. Reporting p-values would help interpret the findings, relevant to the study is not clear.
In discussion section, in line 198 to 258, it demonstrates the treatment was effective at improving sucking, but statistical significance supporting this claim is not demonstrated. The data needs to be analyzed appropriately to make this more comprehensive presentation.
In limitations section, in line 260-270, this content around the small sample size and lack of validated osteopathic scoring are recognize but encourages to add more extra contents for the content of limitations, i.e. lack of blinding in treatment and potential placebo effects. Suggests adding conclusion section at the end of the manuscript.
Comments on the Quality of English LanguageIt needed to have careful editing for minor grammar and style issues for this manuscript.
Author Response
To the Editor and the Reviewers of Heathcare
About the paper “Efficacy of early osteopathic therapy in restoring proper sucking in breastfed infants” by Arianna Parodi, Rosalba Ruffa, Viola De Felice, Marina Sartini,Maria Luisa Cristina, Beatrice Martino, Francesca Bianco, Roberta Di Stefano, Massimo Mazzella
Arianna Parodi, as corresponding author, on behalf of all the authors, has made the requested changes so detailed
Thank you for your comments and suggestions:
In introduction section lacks citations for some statements, such as in line 39-44, the claims that intrauterine position can alter neonatal skull bones and impact breastfeeding. Citations are needed to support these statements, relevant to the study is not clear.
- As suggested a citation about how intrauterine position can alter neonatal skull bones has been added
In method section, in line 71 to 168, it states that an osteopathic score was created for evaluations, but details on how this score was developed, the components assessed, and validation are not provided. More information is needed on this part, relevant to the study is not clear.
- The osteopathic score used in the present paper has been created ad hoc, validation of this score is not in the purpose of the present pilot study. A subsequent investigation on a larger sample of patients could allow to validate the score
In line 90 to 151, no exact duration and data for data collection, relevant to the study is not clear.
- All infants who met the inclusion criteria born between January 2021 and January 2023 were enrolled, with interruptions related to the progress of the covid pandemic in Italy, as added in the Method section
In result section, in line 169 to 197, statistical tests used to compare groups are not reported, so it is unclear if the reported differences are statistically significant. Reporting p-values would help interpret the findings, relevant to the study is not clear.
- Result section has been modified to better explain if the differences found are significant or not.
In discussion section, in line 198 to 258, it demonstrates the treatment was effective at improving sucking, but statistical significance supporting this claim is not demonstrated. The data needs to be analyzed appropriately to make this more comprehensive presentation.
- Discussion section has been modified as suggested to make the presentation more clear
In limitations section, in line 260-270, this content around the small sample size and lack of validated osteopathic scoring are recognize but encourages to add more extra contents for the content of limitations, i.e. lack of blinding in treatment and potential placebo effects. Suggests adding conclusion section at the end of the manuscript.
- Limitation and Conclusion sections have been modified as suggested
Comments on the Quality of English Language: It needed to have careful editing for minor grammar and style issues for this manuscript.
English has been revised
Round 2
Reviewer 2 Report
Comments and Suggestions for Authors
- The authors need to address the data collection procedures and quality control and more details to the ethics considered when conducting this study.
- Also, the discussion required more depth in the interpretation of the findings.
Moderate editing of English language required
Author Response
Thank you for your notes.
About your suggestion:
1) The authors need to address the data collection procedures and quality control and more details to the ethics considered when conducting this study:
Data were collected anonymously on Excel databases. The quality of the data was assessed by two researchers. As far as the ethical issue is concerned, there is the scrutiny of the regional ethics committee both in terms of the type of intervention and in terms of privacy and the storage of sensitive data of our patients
2) Also, the discussion required more depth in the interpretation of the findings.
ok, we tried to better characterize the results of the study and to enhance the possibility of improving the sucking score with a technique that is safe for newborns
Reviewer 3 Report
Comments and Suggestions for Authors
Respected authors,
I have observed significant improvement in the manuscript, but still I feel that there is need for certain improvements:
1) Add citations in the paragraph 2 of the introduction.
2) Still the introduction is very lengthy.
3) Content of line 94-96 seems repetitive, you need to reword them.
4) Figure 2 is not properly explained.
Author Response
Thank you for your notes. Speaking about your suggestion we made the following changes
1) Add citations in the paragraph 2 of the introduction: a citation has been addedd
2) Still the introduction is very lengthy: we sinthesized introduction and a whole sentence has been removed
3) Content of line 94-96 seems repetitive, you need to reword them: sorry, I think you refer to line 84-86. We have changed the sentence to explain better the seconday aim "The secondary aim was to evaluate breastfeeding rates after treatment in the two groups to understand if improving the sucking score also improves the prevalence of breastfeeding"
4) Figure 2 is not properly explained: ok, we tried to better explain the figure that show that sucking score in treated babies is towards normality compared to untreated babies in whom the lack of suction remains and mostly has pathologic values
Reviewer 4 Report
Comments and Suggestions for Authors
Thank you for the opportunity to review the manuscript related to evaluate whether four weeks of osteopathic treatment can normalize the sucking score in a group of neonates with impaired lactation ability in Italy.
A few questions / comments and suggestions:
In line 85-86, the secondary aim must revise more clear, relevant to the study is not clear.
In line 286-290, limitations of the small sample size and lack of validation for the ad hoc osteopathic score are noted. Recommends are made for future study research.
This manuscript indicates as a pilot study, the findings support further research but should not be overgeneralized.
Comments on the Quality of English LanguageMinor editing of English language is required.
Author Response
Thank you very much for your notes
About your suggestion we have made the following changes:
1) In line 85-86, the secondary aim must revise more clear, relevant to the study is not clear: ok, we modified the sentence: The secondary aim was to evaluate breastfeeding rates after treatment in the two groups to understand if improving the sucking score also improves the prevalence of breastfeeding
2) In line 286-290, limitations of the small sample size and lack of validation for the ad hoc osteopathic score are noted. Recommends are made for future study research. This manuscript indicates as a pilot study, the findings support further research but should not be overgeneralized:
ok, thank you for giving us the possibility to clarify our findings better. Eventually a larger number of patients is also needed to better analyze the role of sucking score in breastfeeding performance.